# Implementation of Equilibrium Strategy Aiming at Throughput Maximization of Series Battery Pack

**Rui Cao, Xinhua Liu, Zhengjie Zhang, Mingyue Wang, Hanchao Cheng, Xinlei Gao, Xiaoyu Yan** and **Shichun Yang \***

School of Transportation Science and Engineering, Beihang University, Beijing 100191, China; crcaorui@buaa.edu.cn (R.C.); liuxinhua19@buaa.edu.cn (X.L.); zhengjie_zhang@163.com (Z.Z.); wmy0908@buaa.edu.cn (M.W.); sy2013122@buaa.edu.cn (H.C.); gaoxinlei@buaa.edu.cn (X.G.); yanxiaoyu@buaa.edu.cn (X.Y.)
* Correspondence: yangshichun@buaa.edu.cn

**Abstract:** In the operation process of a power battery pack, the inconsistency among lithium-ion cells may seriously restrict the pack's capacity, power capability and lifetime, which may bring hidden danger to the use of electric vehicles. Equalization management systems (EMSs) are crucial to alleviate such inter-cell inconsistency, whose performance such as accuracy and stability, mainly depends on the setting of equalization control strategies. This paper proposes an equalization strategy aimed at throughput maximization of series battery in the whole life cycle based on Model Prediction Control (MPC). In this paper, a Mean-plus-difference model (M+D) model is selected as the series battery model and the parameters are identified by Recursive Least Squares (RLS). Based on the model predictive control theory, the control model of series battery pack is established and the objective function of maximizing the throughput in the whole life cycle is derived. At the end of the paper, the simulation results show that the proposed equalization strategy can achieve greater life cycle throughput compared with the traditional SOC equalization strategy, which verifies the guiding significance of the equilibrium strategy proposed in this paper.

**Keywords:** BEV (battery electric vehicle); battery management; battery ageing; charge equalization; strategy

## 1. Introduction

Due to the limited power of lithium-ion battery, the power module of electric vehicle is usually composed of multiple batteries in series and parallel to meet the power requirements [1–3]. In the operation process of a power battery pack, the inconsistency among lithium-ion cells may seriously restrict the pack's capacity, power capability and lifetime, which may bring hidden danger to the use of electric vehicles [4,5]. Equalization management systems (EMSs) are crucial to alleviate such inter-cell inconsistency, whose performance such as accuracy and stability, mainly depends on the setting of equalization control strategies [6,7]. To this end, the aim of this paper is to design an equalization strategy aiming at throughput maximization of series battery based on Model Prediction Control (MPC).

To solve the balance control problem of battery pack, the first step is to complete the modelling and parameter identification of battery pack [8,9]. According to the topological structure of the actual connection, the battery pack can be divided into four connection modes: series battery pack, parallel battery pack, series battery pack before parallel battery pack and series battery pack after parallel battery pack [10,11]. A parallel battery pack is often used to analyze the effect of inconsistency between cells on branch shunt, aging and failure of battery pack [12]. The structure of series connection before parallel connection and parallel connection before series connection can be simplified and applied to the parameter analysis of battery [13]. Based on the characteristic that the input current of

cells in the same branch is identical, series batteries can be used in state estimation and equalization management. The equivalent circuit models of the series-connected battery pack consist of Big Cell Model (BCM), Multi-Cell Model (MCM), Vmin-plus-Vmax Model (VVM) and Mean-plus-difference model (M+D). BCM regards the battery pack as a single cell which deduces single cell modelling to the battery pack modelling. MCM uses the same model for each cell of the battery pack and extracts the parameters of each cell to realize the research of the cell characteristics [14]. VVM uses the maximum voltage cell in the charging process and the minimum voltage cell in the discharging process to represent the battery pack [15]; M+D model divides the battery model into a mean value and several deviation correction models to realize the overall description of the battery [16]. Compared with other models, M+D model is simpler in calculation, lower in complexity and higher in identification accuracy, so the M+D model is selected as the series battery model in this paper.

The research in the field of EMS can be divided into two aspects: equalization topology and equalization strategies. Equalization topology is based on the hardware structure and provides the circuit information to the upper strategies. Wu et al. [17] proposed an equilibrium topology based on a Cuk equalizer combined with double-layer selector switch, which led to higher efficiency of energy transfer among batteries. Crespo et al. [18] and Li et al. [19] uses module-level DC/DC power converters in different circuit structure to achieve active equalization of series or parallel battery modules. Xiong et al. [20] proposed a battery equalization circuit on the basis of a bi-directional flyback converter. To sum up, the main aim on equalization topology is to achieve higher efficiency of energy transform.

Equalization strategies can be divided into three categories: variable-based, objective-based and algorithm-based [10]. At present, equalization variables often include operating voltage, SOC, open circuit voltage (OCV), capacity and multivariate fusion [21]. The objective function selects a specific physical quantity to maximize the energy utilization rate and prolongs the service life of the battery [22]. Wang et al. [23] proposed a equalization strategies for series batteries using Fuzzy logic control aimed at shortening the time cost of the equalization process. Song et al. [24] achieved battery pack equalization based on the uniform cell charging voltage curve, and the whole capacity of the pack was improved by more than 10%. Lin et al. [25] used electrochemical-thermal coupling model to realize state of charge (SOC) estimation and achieved pack equalization based on SOC equilibrium. Appropriate equalization algorithm can avoid over equalization and repeated equalization, greatly shorten the equalization time and improve efficiency and robustness [26].

In this paper, an equilibrium strategy aiming at maximum of throughput of the series battery pack is proposed firstly. Figure 1 shows the flow chart of this paper. This work covers battery model selection, parameter identification and equilibrium strategy construction. M+D model is selected as the series battery pack model and the parameters are identified using RLS. The algorithm of this strategy is based on model predictive control. The system model corresponding to model predictive control can be any form of model describing system dynamics [27]. At last, the priority of proposed strategy is proved with a higher throughput by comparison with the traditional SOC equalization.

The rest of this paper consists of the following sections: In Section 2, the parameter identification process for series battery is showed covering the model choosing and the use of algorithm. In Section 3, the equalization strategy is aimed at maximizing the throughput of the battery pack in the whole life cycle. In Section 4, the simulation results verify the superiority of the proposed strategy.

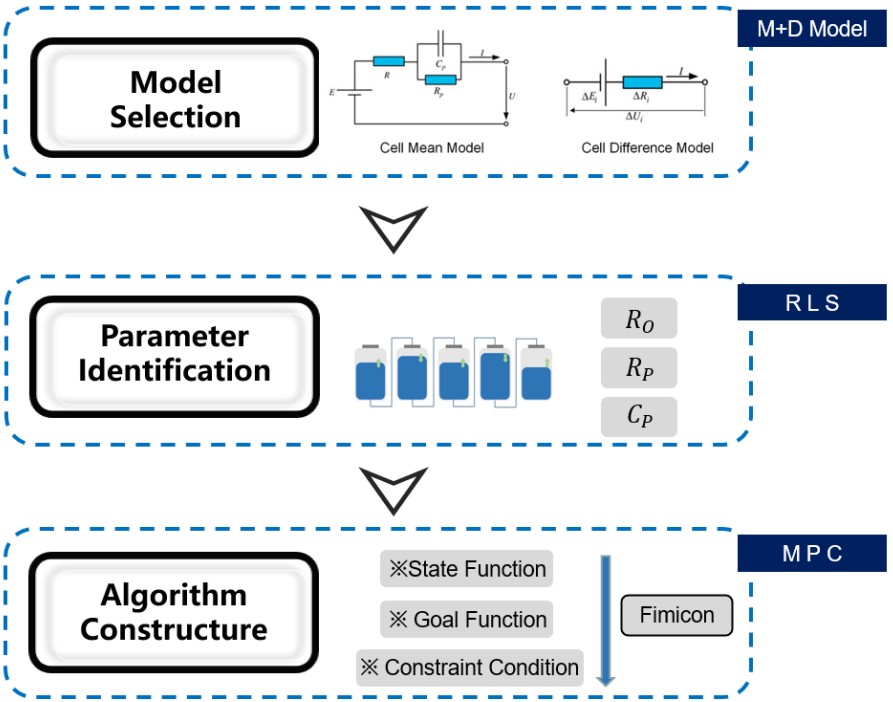

**Figure 1.** Flow chart of this work.

## 2. Parameter Identification for Series Batteries

The M+D model is selected as the model of series battery pack, which can be divided into four kinds according to the different types of difference variables. Based on the selected M+D#2 model, the RLS algorithm is used for parameter identification and the accuracy of the model is tested.

### 2.1. M+D Model

M+D model is divided into cell mean model (CMM) and cell difference model (CDM). The average model represents the overall state of the battery pack and the characteristic parameters can be identified by the average voltage and input current of all the cells of the battery pack. The deviation model describes the difference between the cell and the average value of the battery pack [28]. According to the different types of difference variables, the deviation models are divided into four categories, as shown in Figure 2. CDM#2 is selected as the deviation model for parameter identification which considers resistance and SOC as the difference variables [29]. The state equations of M+D model is shown in Table 1.

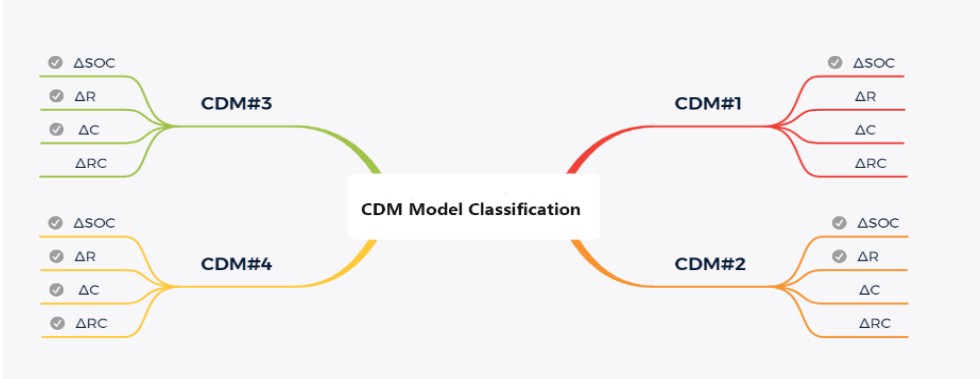

**Figure 2.** CDM model classification.

**Table 1.** State equations of M+D model. Adapted from ref. [28].

| M+D Model | State Equations |
|---|---|
| **CMM**  | $\dot{U}_{P,m} = -\frac{1}{R_{P,m}C_{P,m}}U_{P,m} + \frac{1}{C_{P,m}}I$ <br> $U_{T,m} = U_{OCV,m} - U_{P,m} - R_{O,m}I$ <br> where $U_{T,m}$ represents the mean terminal voltage of the battery pack, $R_{P,m}$ and $C_{P,m}$ are the polarization internal resistance and polarization capacitance. $U_{P,m}$ is the polarization voltage, $R_{0,m}$ represents the ohm internal resistance, and $I$ is the instantaneous current in the circuit. $U_{OCV,m}$ represents the OCV. |
| **CDM#2**  | $\Delta U_{T,i} = \Delta U_{OCV,i}(\Delta SOC_i) - \Delta R_{O,i}I$ <br> where $\Delta R_{0,i}$ represents difference between ohmic resistance and mean ohmic resistance of each cell and $I$ is the instantaneous current in the circuit |

*2.2. Parameter Identification Based on RLS*

Parameters of M+D model are identified with a forgetting factor. To use the RLS algorithm, the CMM is discretized as follows [30].

$$U_{P,m,i+1} = U_{P,m,i}exp(-\Delta t/\tau) + (1 - exp(-\Delta t/\tau)R_{p,m}I_{m,i} \tag{1}$$

$$U_{T,m,i} = U_{OCV,m,i} - U_{m,i} - R_{0,m}I_{m,i} \tag{2}$$

where $\Delta t$ represents sampling interval, $\tau = R_P C_P$, define $E_T = U_T - U_{OCV}$, state function can be rewritten as:

$$\begin{aligned} E_{T,m,i+1} &= exp(-\Delta t/\tau)E_{T,m,i} + (-R_{0,m})I_{m,i+1} \\ &+ (exp(-\Delta t/\tau)R_0 - (1 - exp(-\Delta t/\tau)R_{P,m}I_{m,i} \\ &= \beta_1 E_{T,m,i} + \beta_2 I_{i+1} + \beta_3 I_i \end{aligned} \tag{3}$$

where $\beta_1$, $\beta_2$ and $\beta_3$ are the parameter coefficient of the model.

$$\begin{cases} \beta_1 = exp(-\Delta t/\tau) \\ \beta_2 = -R_0 \\ \beta_3 = exp(-\Delta t/\tau)R_0 - (1 - exp(-\Delta t/\tau))R_{P,m} \end{cases} \tag{4}$$

The parameter $R_{0,m}, R_{P,m}, C_{P,m}$ can be obtained from the above formula.

$$\begin{cases} R_{0,m} = -\beta_2 \\ R_{P,m} = (\beta_1\beta_2 + \beta_3)/(\beta_1 - 1) \\ C_{P,m} = (1 - \beta_1)\Delta t/((\beta_1\beta_2 + \beta_3)\ln(\beta_1)) \end{cases} \tag{5}$$

In order to use RLS algorithm, the equation is written in the form of $y_i = \Phi_i\theta_i$, where

$$\begin{cases} y_i = E_{T,i} \\ \Phi_i = [E_{T,i} \ I_i \ I_{i-1}] \\ \theta_i = [\beta_1 \ \beta_2 \ \beta_3] \end{cases} \tag{6}$$

Similarly, the expression of CDM#2 can be deduced. The RLS identification process is shown in Table 2.

*2.3. Parameter Identification Results*

The OCV-SOC curve of the battery is the average value of the voltage in the process of charging and discharging. The fitting coefficient is obtained by polynomial fitting of the OCV-SOC curve of CMM, of which the results are showed as follows.

$$U_{OCV} = \alpha_1 + \alpha_2 \times x_2 + \alpha_3 \times x_3 + \alpha_4 \times x_4 + \alpha_5 \times x_5 + \alpha_6 \times x_6 + \alpha_7 \times x_7 \tag{7}$$

$$\begin{cases} \alpha_1 = -85.2710 \\ \alpha_2 = 299.0915 \\ \alpha_3 = -425.5480 \\ \alpha_4 = 312.0505 \\ \alpha_5 = -118.0295 \\ \alpha_6 = 25.3155 \\ \alpha_7 = 15.3911 \end{cases}$$

Based on the experimental data of HPPC experiment, the parameters of series battery pack model are identified. The identification results are shown in Table 3.

**Table 2.** M+D Model parameter identification.

CMM:
(i) Initialization: $\phi_m$, $\theta_m$, $K_m$, $P_m$, $\lambda$

(ii) Calculate and measure the mean voltage: $U_{T,m,i} = \sum\limits_{k=1}^{n} U_{T,k,i} / n$

(iii) Calculate of mean cell gain matrix: $K_{m,i} = \left( P_{m,i-1}\phi_{m,i}^{T} \right) / \left( \lambda + \phi_{m,i}^{T} P_{i-1}\phi_{m,i} \right)$

(iv) Calculate the mean cell error covariance matrix: $P_{m,i} = \left( P_{m,i-1} - K_{m,i}\phi_{m,i}^{T} P_{m,i-1} \right) / \lambda$

(v) Update mean cell parameter matrix: $\theta_{m,i} = \theta_{m,i-1} + K_{m,i}\left( E_{T,m,i} - \phi_{m,i-1}^{T}\theta_{m,i-1} \right)$
(vi) Update estimated voltage:

CDM#2:
$U_{T,m,i} = \phi_{m,i}^{T}\theta_{m,i} + U_{OCV,m,i} \ U_{T,k,i} = U_{T,m,i} + \Delta U_{OCV,k,i} + \Delta R_{0,k} I_i \ U_{T,i} = \sum\limits_{k-1}^{n} U_{T,k,i}$
where $U_{T,k,i}$ represents the terminal voltage value of the cell $k$ at sampling point $i$, $U_{T,i}$ represents the terminal voltage value of the battery pack at time $i$.
In addition
$E_{T,i} = U_{T,i} + U_{OCV,N,i} \ E_{T,k,i} = U_{T,k,i} + U_{OCV,k,i} \ E_{T,\max,i} = U_{T,\max,i} + U_{OCV,\max,i}$
$E_{T,\min,i} = U_{T,\min,i} + U_{OCV,\min,i} \ E_{T,m,i} = U_{T,m,i} + U_{OCV,m,i}$
The forgetting factor $\lambda = 0.95$

**Table 3.** The identification results.

| Parameter | Cell #1 | Cell #2 | Cell #3 | Cell #4 | Cell #5 | CMM |
|---|---|---|---|---|---|---|
| $R_O$ (m$\Omega$) | 1.3626 | 1.3652 | 1.3679 | 1.3706 | 1.3701 | 1.3676 |
| $R_P$ (m$\Omega$) | | | 0.6017 | | | |
| $C_P$ ($10^4$ F) | | | 1.1969 | | | |

The accuracy of the identification results is verified, as shown in the Figure 3. The fitting error is less than 1%, of which the accuracy meets the requirements.

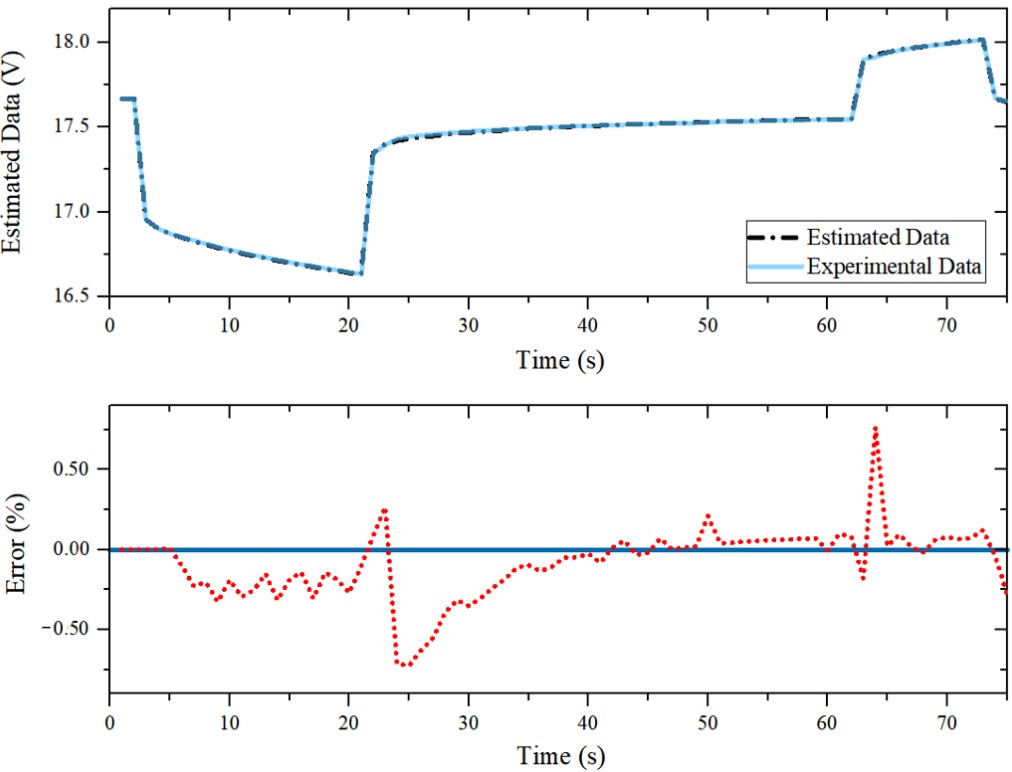

**Figure 3.** Identification results.

## 3. Equalization Strategy Design Based on Model Predictive Control

Based on the results of parameter identification, the MPC-based equalization strategy is designed. The derivation of MPC space equation, the determination of objective function, the determination of constraint conditions and the solution of quadratic problem are completed respectively [31,32]. The result is finally obtained using the fimicon algorithm in Matlab and the simulation results are tested by comparative verification.

### 3.1. State Space Model of Series Batteries

In order to explain the effect of the equalization strategy and take into account the computational complexity, the number of series batteries is selected as five. Both the prediction time domain p and control time domain m are selected as six. A complete multi-input multi-output state space model is obtained by taking the first-order RC equivalent circuit model of battery as the prediction model [33,34].

Cell equivalent circuit model is:

$$x(k+1) = A_1 x(k) + B_1 I(k) \tag{8}$$

$$y(k) = C_1 x(k) + D_1 I(k) + U_{oc}(k) \tag{9}$$

where the state variable is: $x(k) = [U_1(k) \quad SOC(k)]^T$. The output variable is: $y(k) = U_t(k)$.

The four coefficient matrices are:

$$A_1 = \begin{bmatrix} e^{-\frac{\Delta t}{R_p C_p}} & 0 \\ 0 & 1 \end{bmatrix} B_1 = \begin{bmatrix} R_1 \times \left(1 - e^{-\frac{\Delta t}{R_p C_p}}\right) \\ -\frac{\eta \Delta t}{C_{bat}} \end{bmatrix}$$

$$C_1 = \begin{bmatrix} -1 & 0 \end{bmatrix} D_1 = -R_0$$

Based on the model predictive control theory, the space equation is derived:

$$X_p(k+1|k) = S_x x(k) + S_{ux} U(k) \tag{10}$$

$$Y_p(k+1|k) = S_y x(k) + S_{uy} U(k) \tag{11}$$

According to the selected parameters: number of batteries, prediction time domain and control time domain, it can be derived as follows:

$$X_p(k+1|k) = \begin{bmatrix} x_{1,1}(k+1|k) \\ \vdots \\ x_{5,1}(k+1|k) \\ x_{1,2}(k+2|k) \\ \vdots \\ x_{5,6}(k+6|k) \end{bmatrix}_{60\times1} \quad Y_p(k+1|k) = \begin{bmatrix} y_{1,1}(k+1|k) \\ \vdots \\ y_{5,1}(k+1|k) \\ y_{1,2}(k+2|k) \\ \vdots \\ y_{5,6}(k+6|k) \end{bmatrix}_{30\times1} \quad x(k) = \begin{bmatrix} x_1(k) \\ x_2(k) \\ \vdots \\ x_5(k) \end{bmatrix}_{10\times1}$$

$$U(k+1|k) = \begin{bmatrix} I_{1,1}(k+1|k) \\ \vdots \\ I_{1,6}(k+1|k) \\ I_{2,1}(k+2|k) \\ \vdots \\ I_{5,6}(k+4|k) \end{bmatrix}_{30\times1} \quad S_x = \begin{bmatrix} A \\ A^2 \\ A^3 \\ A^4 \\ A^5 \\ A^6 \end{bmatrix}_{60\times10} \quad S_y = \begin{bmatrix} CA \\ \sum_{i=1}^{2} CA^i \\ \vdots \\ \sum_{i=1}^{5} CA^i \\ \sum_{i=1}^{6} CA^i \end{bmatrix}_{30\times10} \quad S_{ux} = \begin{bmatrix} B & 0 & \cdots & 0 \\ AB & \ddots & & 0 \\ \vdots & & \ddots & 0 \\ A^5B & A^4B & \cdots & B \end{bmatrix}_{60\times30}$$

$$S_{uy} = \begin{bmatrix} CB+D & 0 & \cdots & 0 \\ CAB & \ddots & & 0 \\ \vdots & & \ddots & \vdots \\ CA^5B & CA^4B & \cdots & CB+D \end{bmatrix}_{30\times30}$$

where two corner markers in the lower right corner of a letter $x_{ij}$, indicating the state of the $i$th battery in the battery pack at the j moment. The matrix is a diagonal matrix formed by the coefficient matrix in the cell model, which can be written as:

$$\begin{aligned} A &= diag(\ A_1 \quad A_2 \quad A_3 \quad A_4 \quad A_5\ ) \\ B &= diag(\ B_1 \quad B_2 \quad B_3 \quad B_4 \quad B_5\ ) \\ C &= diag(\ C_1 \quad C_2 \quad C_3 \quad C_4 \quad C_5\ ) \\ D &= diag(\ D_1 \quad D_2 \quad D_3 \quad D_4 \quad D_5\ ) \end{aligned}$$

### 3.2. Objective Function and Solution Method of Equilibrium Strategy

① The objective function of maximizing the throughput.

Define $c = C_{act}/C_{init}$, where $C_{init}$ is the initial capacity of the battery, which can be regarded as a constant, define $K = dc/dn$ as the battery aging rate, define $n$ as the number of life cycles, define at the time that $c = 0.8$, the number of life cycles as the maximum number of battery cycles, $Q$ represents the throughput of the battery life cycle [3]. Where

$$Q = C_{init} \int_0^N C dn \tag{12}$$

$$C = 1 - \int_0^n K dn \tag{13}$$

Combine the above equations:

$$Q = C_{init} \int_0^N (1 - \int_0^n K dn) dn \tag{14}$$

It is obvious that the larger the value $K$ is, the smaller the value $N$ is, the smaller the battery throughput is. Conversely, the smaller the value $K$ is, the larger the battery throughput is. Therefore, it can be concluded that there is a positive correlation between the battery life cycle throughput and the aging rate [3], the empirical formula of aging rate and aging rate was put forward through data fitting

$$K(SOC) = a_1 SOC^2 + a_4 DOD \tag{15}$$

$$a_1 = 6.1852e^{-4} a_2 = 3.911e^{-6}$$

According to the above formula, the equilibrium objective function can be written as follows:

$$Jmin = \sum_{i=1}^{5} \sum_{j=1}^{6} K_{i,j min} \tag{16}$$

where $K_{i,j}(SOC)$ represents the battery aging rate of ith cell in the pack at j moment.

② The objective function of the traditional *SOC* equalization

In order to prove the effectiveness of the life cycle balancing strategy, the balancing strategy considering *SOC* consistency is used as a comparative analysis. *SOC* consistency is to make the *SOC* of each cell in the whole battery reach the average *SOC*, so the objective function can be obtained as follows:

$$Jmin = \sum_{i=1}^{5} \sum_{j=1}^{6} \left( SOC_j(i) - \overline{SOC_j} \right)_{min} \tag{17}$$

where $SOC_j(i)$ represents the *SOC* of the ith cell in the pack at j moment, $\overline{SOC_j}$ represents the mean *SOC* of the whole pack.

③ Input constraints

For the model predictive control of this problem, the input constraints are as follows:

$$SOC_{min} < SOC(k) < SOC_{max}$$

$$U_{min} < U(k) < U_{max}$$

$$Y_{p,min} < Y_p(k) < Y_{p,max}$$

$$\Delta U_{min} < \Delta U(k) < \Delta U_{max} \tag{18}$$

$$Y_p(1:5)^T U(1:5) = 0$$

$$Y_p(6:10)^T U(6:10) = 0$$

$$Y_p(11:15)^T U(11:15) = 0$$

$$Y_p(16:20)^T U(16:20) = 0$$

$$Y_p(21:25)^T U(21:25) = 0$$

$$Y_p(26:30)^T U(26:30) = 0$$

The first term of the above formulas is set to limit the *SOC*. The maximum aging rate of the battery is at the end of *SOC* 0 or 1 and the function of which is a convex function in the interval, so that the *SOC* is controlled in the range [0.1, 0.9]. The second term is set to limit the output voltage of the battery and the output voltage range of the battery used in this work is [2.7 V, 4.2 V]. The third equation is the limitation of the equalization current, in which the maximum equalization current is not more than 10 A. The fourth is the fluctuation limit of the equalization voltage. During the equalization process, the batteries charge and discharge each other. From the overall point of view of the battery pack, the equalization power is zero. As a result, the power balance constraint is added, which is five to eight equations of the above formulas. Based on the goal function and constraint condition, the result can be obtained using the fimicon optimization function in Matlab in time-varying condition.

Since the objective function of this paper has the compensation term of DOD causing certain nonlinearity, fmincon function is applied to solve the problem [35,36]. In this way, the optimal control sequence is obtained by invoking the fmincon function at each time and the first element of the solution is applied to the system, of which the process is repeated so as to achieve the equilibrium control under different objectives.

## 4. Simulation Result

This article compares the results of the two equalization strategies under static and constant current charging conditions, of which the simulation results verify that the equalization strategy proposed in this article can maximize the throughput of the battery pack throughout its life cycle. In the following analysis, the balance strategy for maximizing throughput throughout the life cycle is option 1 and the traditional SOC uniform balance strategy of the battery pack is option 2.

In the static state, two equalization strategies are used to balance the battery pack. Figure 4a,b show the results of the two equalization strategies. Figure 4c shows the comparison of the instantaneous aging rate under the two strategies. It can be seen that the sum of instantaneous aging rate of option 1 is smaller and the battery pack's life cycle throughput is greater.

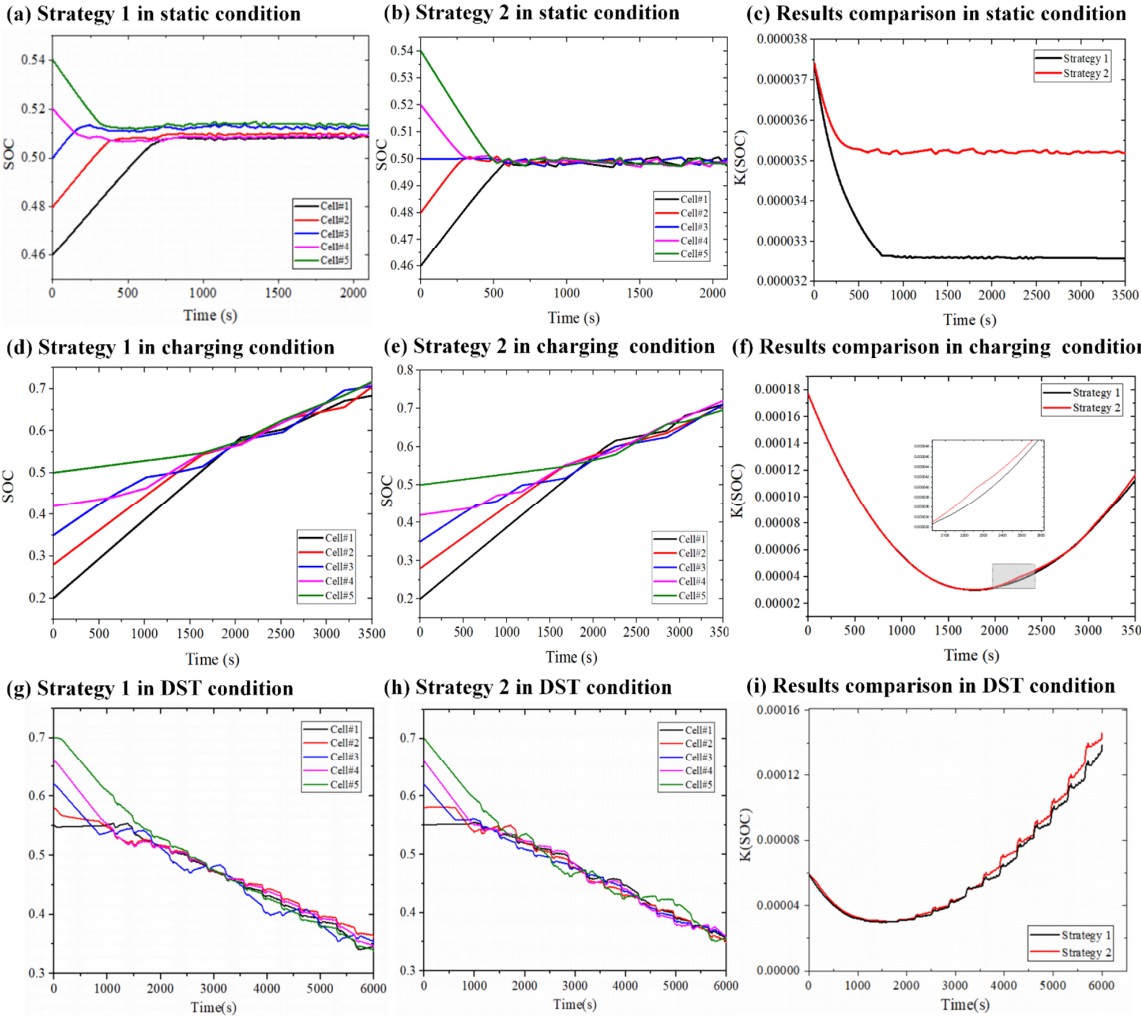

**Figure 4.** Simulation results.

Figure 4d,e show the simulation results of the two strategies in the constant current charging condition. In Figure 4f, the result of the sum of instantaneous aging rate shows that option 1 can achieve a little greater life cycle throughput of the battery pack.

Figure 4g,h show the simulation results of the two strategies in the DST condition. In Figure 4i, the result of the sum of instantaneous aging rate shows that option 1 can achieve a greater life cycle throughput of the battery pack.

The comparison results prove that the equilibrium strategy proposed in this paper can achieve greater life cycle throughput of the series battery pack than the traditional SOC equilibrium strategy. This equilibrium strategy combines the battery aging state and MPC algorithm to achieve equation of battery pack in different conditions. Moreover, the proposed equilibrium strategy as an alternative of previous strategy provides a possibility to improve BMS performance to some extent.

## 5. Conclusions

This paper proposed an equilibrium strategy based on model predictive control aimed at maximum throughput in battery whole life cycle. This work contains battery pack model selection, parameters identification and equilibrium strategy construction, which achieved closed loop simulation. The identified model in this work can achieve an accurate description of the series battery pack considering the mean and difference value in a time-varying condition. The predictive control model is selected as the main algorithm of the equalization strategy, which can adapt to various actual conditions. The battery series used in this work consists of five cells. For further applications, this equalization strategy is adequate for series battery pack containing more cells and can realize a higher profit of throughput of the battery pack. As a result, the equalization strategy proposed in this work can be obtained in the context of real-world battery management easily. Compared to the traditional battery equalization strategy, this work, aiming at throughput maximum, considered not only voltage and SOC, but also the ageing state of the battery cell, and can achieve the internal balance of the battery pack safely and efficiently.

**Author Contributions:** Conceptualization, R.C. and Z.Z.; methodology, R.C.; software, R.C. and X.L.; validation, H.C., R.C. and M.W.; formal analysis, X.G.; investigation, S.Y.; resources, S.Y.; data curation, X.L.; writing—original draft preparation, R.C.; writing—review and editing, X.L. and X.Y.; visualization, Z.Z.; supervision, H.C.; project administration, M.W.; funding acquisition, S.Y. All authors have read and agreed to the published version of the manuscript.

**Funding:** This work was supported by the Xinjiang Uygur Autonomous Region Regional Collaborative Innovation Program (Science and Technology Assistance to Xinjiang Project: No.2019E0240), the National Key Research and Development Program of China (2018YFB0104400), and National Natural Science Foundation of China (No. U1864213).

**Institutional Review Board Statement:** Not applicable.

**Informed Consent Statement:** Not applicable.

**Data Availability Statement:** The [hppc.xls] data used to support the findings of this study are available from the corresponding author upon request.

**Acknowledgments:** The authors would like to thank Evs34 conference group for recommendation.

**Conflicts of Interest:** The authors declare no conflict of interest.

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
