# Peer review of "Implementation of Equilibrium Strategy Aiming at Throughput Maximization of Series Battery Pack"

_wevj, doi:10.3390/wevj12040208_

Round 1
Reviewer 1 Report
This work presents an equalization strategy aiming at throughput maximization of series battery in the whole life cycle based on Model Prediction Control.
The manuscript is interesting and the reported simulation results and findings are good.
However, the Authors should refine the maths making the symbols more consistent.
Table 2 should be better organized in a pseudocode format.
It would be interesting to analyse the effects of the variation (increase) of the number of batteries in series (now 5).
Reviewer 2 Report
Nowadays, the battery technology is an important task. Moreover, the methods of increasing efficiency are welcome. The authors should improve the Introduction section. The novelty of the paper cannot be proved without comparing the proposed method with the similar in this field. Recent references (2021) should be introduced. The authors should improve the research method by adding a flowchart of the implementation of the proposed method. The method of obtaining the minimum of the objective function is not described. The auhors should clarify the steps of obtaining the solutionAuthor Response
Please see the attachment.

Round 2
Reviewer 1 Report
This article has been slightly improved and the authors have satisfied my suggestions quite well.
Reviewer 2 Report
The paper has been improved. Please, revise the entire text of the manuscript (i.e the fimicon optimization, page 8)